## Rapid Communication

spurs and grooves; coral island stability; sediment transport; coral reefs

**Corresponding author:**
Ana Vila-Concejo;
Email: ana.vilaconcejo@sydney.edu.au

# Grooves in forereefs act as transport channels to deliver coral rubble during tropical cyclones

Ana Vila-Concejo[1,2] , Lachlan A. Perris[1,2], Ana Paula da Silva[1,2], Kate Whitton[1,2], Liav Meoded-Stern[1,2], Wan-Yi Steilberg-Liu[1,2], Ruby Holmes[3], Heinrich Breuer[3], Maria Byrne[2,4], Thomas E. Fellowes[1,2,5], Tristan Salles[1,2], Bradley D. Morris[6] and Eleanor Bruce[1,2]

[1]Geocoastal Research Group, School of Geosciences, The University of Sydney Faculty of Science, Australia; [2]Marine Studies Institute, The University of Sydney Faculty of Science, Australia; [3]One Tree Island Research Station, The University of Sydney Faculty of Science, Australia; [4]Integrative Marine Biology Group, School of Life and Environmental Sciences, The University of Sydney Faculty of Science, Australia; [5]Water Research Laboratory, School of Civil and Environmental Engineering, UNSW Sydney, Manly Vale, NSW 2093, Australia and [6]Water, Wetlands and Coasts Science, Science and Insights Division, Department of Climate Change, Energy, the Environment and Water, NSW Government, Sydney, Australia

## Abstract

The long-term stability of coral reef islands and associated reef top sedimentary landforms requires the delivery of sediment from the forereef, but the rates and pathways of sediment delivery to these systems remain unclear. Spurs and grooves (SAGs) are ubiquitous geomorphic features fringing coral reefs, characterised by shore-normal coral ridges (spurs) separated by channels (grooves) with either bare substrate or a relatively low sediment infill. SAGs dissipate wave energy, facilitate offshore sediment transport and enhance nutrient exchange. Here we present the first evidence that SAG can also act as channels for onshore transport of rubble during high-energy events, contributing to maintaining reef islands and rubble-based ecosystems.

## Impact statement

This study documents a rare and consequential sediment transport event following a sequence of climate-related disturbances on the Great Barrier Reef. In the early months (Austral summer) of 2024, the reef experienced the ninth global bleaching event, with catastrophic impacts at One Tree Reef, a site of long-standing significance in Geoscience. One year later, in March 2025, Tropical Cyclone Alfred remobilised the resulting rubble and transported it onto the reef flat. While grooves (the channels of spur and groove systems) have long been suspected to play a role in sediment dynamics, the prevailing view has been that it is "impossible" for rubble to move upslope through them to the reef flat. Our study shows the first evidence that large volumes of coral rubble are delivered to the reef flat through these grooves. We present qualitative and quantitative drone surveys and field measurements of topography and bathymetry to document this process. These findings carry implications for sediment budgets, interpretations of past reef-building processes and forecasts of reef and shingle island evolution under climate change. These insights are especially timely given their relevance to increasingly used restoration strategies (*e.g.*, rubble stabilisation) and to global sustainability efforts, particularly United Nations Sustainable Development Goals related to Small Island Developing States, such as SDG13 (Climate action), SDG14 (Life below water) and SDG 11 (Sustainable cities and communities). These insights are important given the relevance to millions of people globally whose livelihoods are inextricably linked to coral reef ecosystems.

## Introduction

Coral reef islands are dynamic environments that, by adjusting to the environmental conditions (Masselink et al., 2020), remain among the most climate-vulnerable ecosystems on Earth. Physical modelling of gravel islands has demonstrated that islands could build up and adapt to sea level rise (Tuck et al., 2019; Masselink et al., 2020). However, the pathways and rates of sediment supply, without which build-up cannot happen, are event-driven and remain poorly constrained (de Bakker et al., 2025). Islands composed of biogenic carbonate materials are built from sediments derived from the surrounding forereefs and reef flats. Consequently, the sediment supply is threatened by climate change disruptions such as increased marine heatwaves and intense tropical cyclones, which reduce overall reef productivity (de Bakker et al., 2025).

Here, we present new evidence of the pathways of sediment delivery from a bleached coral reef during a tropical storm.

Spurs and grooves (SAGs) act as living breakwaters (Munk and Sargent, 1948) on the forereef of coral reefs. Spurs are parallel ridges of living carbonate material (coral and algae) separated by regularly spaced channels (grooves) (Duce et al., 2016). SAGs are usually oriented perpendicular to the reef crest and/or the incoming waves. They form a "comb-tooth" pattern (6–8 m wide) extending from the reef crest to depths of 20 m (Gischler, 2010). SAGs show considerable variations in morphology predominantly driven by the prevailing hydrodynamic energy (Duce et al., 2020). SAG morphology has been documented across different reef morphologies, including fringing reefs, barrier reefs and atolls through coral-rich seas (Duce et al., 2016). While they have been studied since the mid twentieth century (Munk and Sargent, 1948), recent years have seen an increased focus on SAG research, including remote sensing (Duce et al., 2016), numerical modelling (Rogers et al., 2013; da Silva et al., 2020; Watanabe et al., 2023; Perris et al., 2024) and field measurements (Storlazzi et al., 2004; Monismith et al., 2013; Rogers et al., 2015; Duce et al., 2020, 2022; Acevedo-Ramirez et al., 2021; Sartori et al., 2025). These studies have confirmed that SAG morphology and hydrodynamics are linked to wave climate, from incident short waves to infragravity. They are important wave dissipaters protecting the areas behind reefs (Monismith et al., 2013; Duce et al., 2022; Watanabe et al., 2023; Perris et al., 2024).

SAGs exist under a broad range of wave energy settings and associated water flows (Duce et al., 2016, 2020). It is mostly through modelling that water flows in the SAG zone have been investigated, with research indicating the existence of counter-rotating Lagrangian circulation cells in the grooves (Rogers et al., 2013; da Silva et al., 2020). However, it is unclear whether the flows that occur in the grooves drive sediment transport onshore (towards the reef flat) or offshore. While numerical modelling studies (Rogers et al., 2013) mostly found gentle offshore flow over the shallow part of the grooves, others (da Silva et al., 2020) showed a gentle onshore current in most of their simulations. More recently, field measurements have shown offshore flow in shallow grooves under low wave energy conditions (Perris et al., 2025; Sartori et al., 2025). While Rogers (2013) posits that under strong wave forcing, the offshore transport on the grooves would be reduced or potentially reversed, such conditions have never been measured or observed. A study using tracers of the effects of Category 4 Typhoon Robyn in 1993 in Kume Island (Ryukyus, Southern Japan) showed that coral clasts could be transported from the forereef (up to 12 m depth) onto the reef flat, but those clasts were not located on the grooves (Kan, 1995). Other indications of onshore transport result from personal observations by the authors of the current manuscript of onshore imbrication of coral clasts witnessed while deploying instruments in the Maldives and the Great Barrier Reef.

One Tree Reef (OTR) is a wave-exposed mesotidal platform reef located on the southern Great Barrier Reef (GBR, Figure 1). Its eastern margin comprises a rubble-dominated reef flat containing an estimated 14 million tons of rubble derived from the reef front (Thornborough and Davies, 2011). It is one of the few reefs in the GBR containing a shingle island on its exposed margin. Coral rubble is the generic term used to denote sediments resulting from the fragmentation of calcifying organisms, including coral and molluscs. It refers to sediment larger than sand (>2 mm) and typically up to

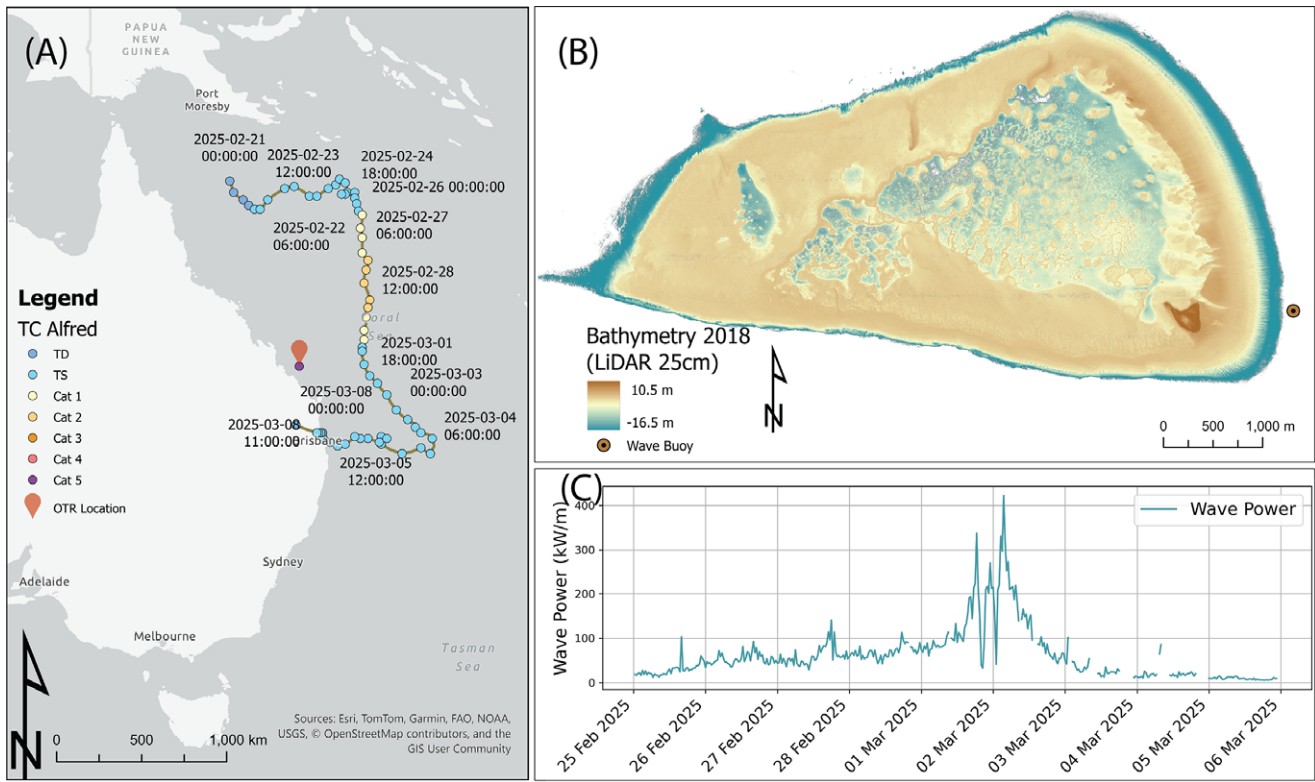

**Figure 1.** (A) Location of One Tree Island on the southern Great Barrier Reef off the coast of NE Australia and track and Saffir-Simpson category for Tropical Cyclone Alfred in March 2025, note in the legend, TD= Tropical Depression and TS= Tropical Storm (Source IBTrACS (Knapp et al. 2010)); (B) LiDAR Digital Elevation Model (Vertical datum is mean sea level, MSL) with 25 cm resolution of One Tree Reef (OTR) measured in 2018 (Harris et al. 2023) showing the location of the Spotter wave buoy; (C) Wave power from the southern One Tree Island Research Station (OTIRS) wave buoy on the east flank of One Tree Reef (see B for location) during TC Alfred.

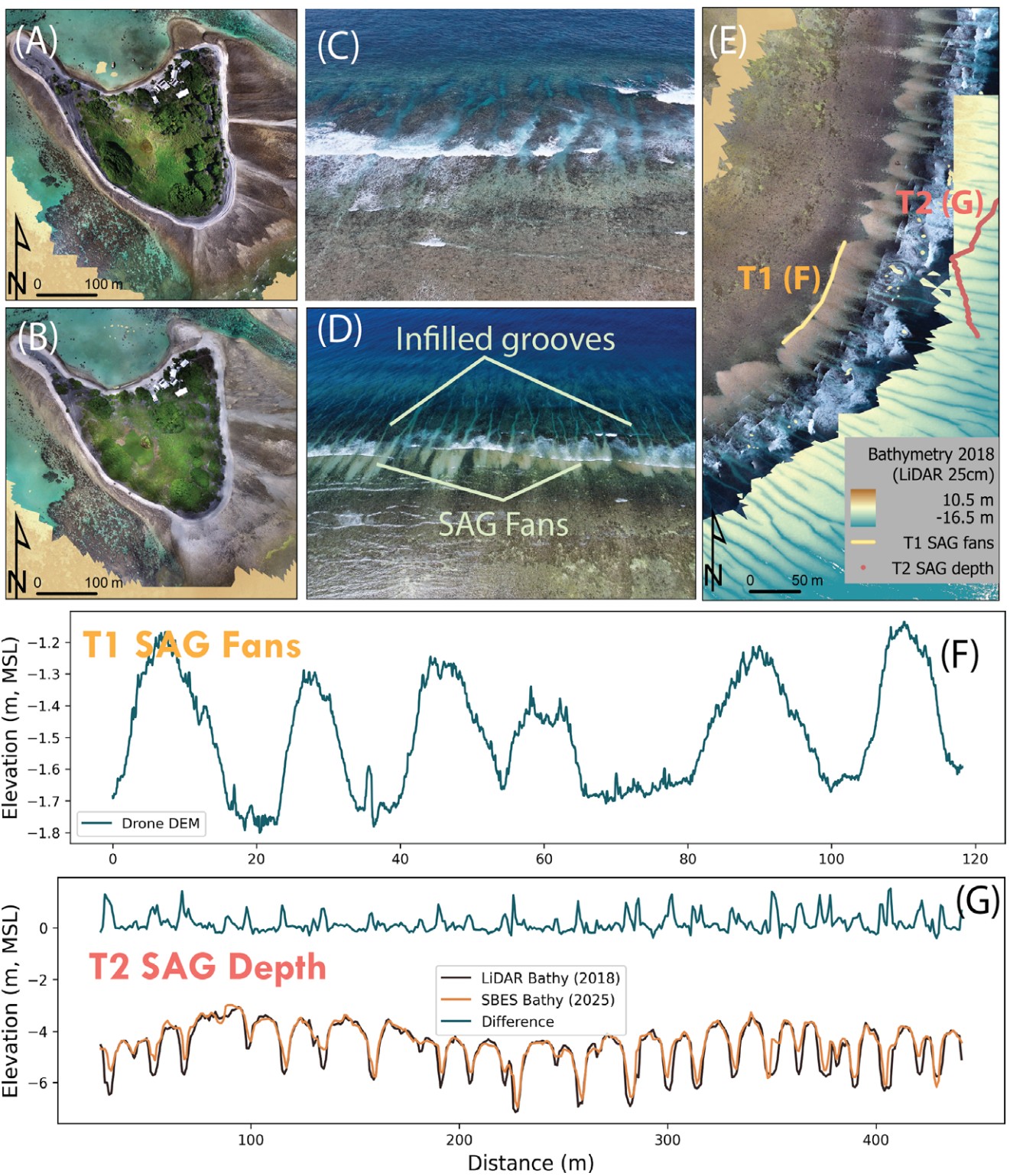

**Figure 2.** Orthomosaics of One Tree Island with adjacent rubble reef flat in November 2022 (A) and post TC Alfred in March 2025 (B); the grooves off the SE of One Tree Reef are clear of sediment in November 2024 (photo by Lachlan Perris, C) and infilled with coral rubble fans spilling sediment as SAG Fans onto the reef flat post TC Alfred in March 2025 (photo by One Tree Island Research Station, OTIRS, D); Orthomosaic of the SAG at One Tree Reef post TC Alfred (March 2025), showing the sediment fans delivered through the groove channels, and the locations of T1 and T2, the SAG bathymetry corresponds to data obtained with LiDAR in 2018(E); T1 shows a cross-section of the edges of the rubble fans extracted from the drone data (F); and, T2 demonstrates infilling of the grooves by comparing a LiDAR bathymetry from 2018 with a single beam bathymetry (T2) measured in 2025 post TC Alfred (G).

boulder size in the Udden-Wentworth scale (>256 mm), with a variety of morphologies ranging from branching to tabular (Rasser and Riegl, 2002). The coral rubble is dumped on the reef flat during a large, high-energy event and is then reworked (Thornborough and Davies, 2011) with some of it forming rubble spits or tracts (Shannon et al., 2013) that deliver rubble to the shingle island (Talavera et al., 2021). The SAGs on the eastern forereef of OTR (Figure 2) have spurs with ~90% coral cover and grooves with a "U"-shaped cross-section,

and at the base have a variable amount of large rubble clasts with occasional pockets of coarse sand (Duce et al., 2016). Rubble environments are ubiquitous in coral reefs (Blanchon et al., 1997; Thornborough and Davies, 2011), and previous ecosystem restoration studies have highlighted that rubble mobility hinders coral growth (Ceccarelli et al., 2020). On the other hand, recent studies suggest that rubble fields are important ecosystems that serve as potential targets for coral settlement (Heyward et al., 2024).

In February–March 2024, OTR underwent the worst bleaching in its history, causing up to 53% coral mortality (Byrne et al., 2025), more acute in areas of low hydrodynamic energy (Meoded-Stern et al., 2025). This event generated large amounts of coral rubble resulting from the dead coral collapse. One year after the bleaching, on 1 March 2025, Tropical Cyclone (TC) Alfred passed 340 km east of One Tree Island (Figure 1A). It was a slow-moving Category 1–2 cyclone (Saffir-Simpson Scale) that downgraded to Tropical Storm later that day, still sustaining strong winds. Our wave Spotter Buoy located on the SE off the reef (Figure 1B) recorded Significant Wave Heights ($H_s$) up to 7 m with associated Peak Periods ($T_p$) of 12 s. This led to wave power exceeding 250 kW/m sustained over nearly 12 h (Figure 1C). Fortunately, for One Tree Island Research Station, the largest waves occurred during the astronomical low tides.

As expected, TC Alfred delivered large volumes of rubble to the reef flat, modifying the overall morphology of the island (Figure 2A,B; see white rubble in 2B over the reef flat and on the northern tip). The mechanisms acting during these event-driven stochastic occurrences are not well established/studied, and scientists have debated for years whether SAGs could act as channels to deliver sediments to the reef flat under high-energy conditions. In March 2025, immediately after the passage of TC Alfred, we captured a photo showing grooves that are normally relatively empty of sediment (Figure 2C) were completely infilled with sediment and had newly formed sedimentary fans of rubble extending out from each groove on the eastern margin of OTR (Figure 2D). This is the first evidence showing the grooves acting as channels for rubble delivery onto the reef flat from decades of SAG research worldwide. Measurements from drone surveys post-Alfred show the distinct one-to-one spatial alignment between the rubble fans and the grooves, with each groove producing a single fan (Figure 2E). Bathymetric measurements show over 1 m of rubble infilling on the previously mostly bare grooves (Figure 2F).

Our observations and measurements establish, for the first time, grooves as channels transporting coral rubble onto the reef flat during high-energy conditions. This contrasts with the commonly accepted paradigm that rubble on the reef flats is transported from the spurs during high-energy conditions, and that rubble in the grooves remains trapped there or, under the right conditions, moves down the groove (Kan et al., 1997; Hubbard and Dullo, 2016; Duce et al., 2020; Sartori et al., 2025). The observed groove onshore transport occurred during extreme high-energy conditions, with TC Alfred being in the top 99th percentile of TC-generated waves for the study area. The threshold at which this onshore transport initiates is likely location dependent and deserves further investigation, as onshore rubble transport is crucial for the persistence of rubble islands, and TCs are expected to become more intense and less frequent in the Southern Hemisphere (Knutson et al., 2020). Finally, our observations highlight that the function of SAGs is not limited to the dissipation of wave energy but enables onshore rubble transport and contributes to coral environment dynamics essential for reef ecosystem health. While we have shown that rubble delivery plays an important role in the island stability of shingle islands, it has also been shown to influence sandy islands such as the Maldives (Gea-Neuhaus et al., 2025). Reef building is more than coral growth; it is a complex interplay of carbonate production, destruction and transport, as well as the reincorporation of sediment into the reef framework (Hubbard and Dullo, 2016). In the face of a changing climate, there is an urgent need to understand natural coral reef systems (Streit et al., 2024). Understanding rubble dynamics must precede rubble restoration as a management tool in natural systems.

**Open peer review.** To view the open peer review materials for this article, please visit http://doi.org/10.1017/cft.2025.10019.

**Data availability statement.** The data will be made available once processed and published in its entirety. In the meantime, colleagues can address data requests to the corresponding author.

**Acknowledgements.** The authors would like to acknowledge the First Nation peoples on whose land the One Tree Island Research Station stands, the Bailai, Gurang, Gooreng and Taribelang Bunda (FNBGGGTB) People. The authors would also like to acknowledge the Gadigal People from the Eora Nation, where the University of Sydney stands. This research has been partially funded by the Australian Research Council (ARC) Discovery Project DP220101125 and the Marine Resource Initiative (MRI) project with Geoscience Australia and the Australian Department of Foreign Affairs and Trade. Wave data from the spotter buoy was processed by the Coastal and Marine Science team, Department of Climate Change, Energy, the Environment and Water (DCCEEW), NSW Government, Australia. This work represents a contribution to the ARISE project (UKRI grant EP/X029506/1). We thank Jody Webster and the MARS5007 2025 cohort for supporting some of the measurements. Claudia Le Quesne and Lara Talavera contributed to the data processing for the November 2022 dataset.

**Author contribution.** AV-C led the research, processed data and wrote the manuscript. LAP, KW, W-YS-L, RH and HB obtained and processed field data. BDM processed the Spotter Buoy wave data. APdS, LM-S, MB, TEF, TS and EB contributed to data analysis. All authors contributed to writing and editing the manuscript.

**Financial support.** This research has been partially funded by the Australian Research Council (ARC) Discovery Project DP220101125 and the Marine Resource Initiative (MRI) project with Geoscience Australia and the Australian Department of Foreign Affairs and Trade. LAP was supported by an RTP scholarship. KW and APdS were supported by DP220101125.

**Competing interests.** The authors declare none.

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
