## [Reviewer Report]

This is an extremely interesting and novel piece of work that should be published. It details for the first time field evidence of the role of spurs and grooves in delivering sediment to the reef flats during cyclones. This finding has implications for understanding the processes of sediment delivery to reef flats and ultimately eventually to islands. Some minor suggested amendments are provided below.

Suggest to reference Tuck et al. 2019 in reference to physical models – line 59. (Megan E. Tuck, Paul S. Kench, Murray R. Ford, Gerd Masselink; Physical modelling of the response of reef islands to sea-level rise. Geology 2019;; 47 (9): 803–806. doi: https://doi.org/10.1130/G46362.1)

Line 62 – suggest change ‘forereefs’ to ‘reefs’ or ‘forereefs and reef flats’ as it is not just the forereefs that produce sediment.

Line 63 – be specific about climate change disruptions – increased sea surface temps, and maybe increased intensity of tropical cyclones?

Line 86 – are these circulation cells located in grooves?

Line 95 – please indicate where this study was undertaken

Line 98 – do you mean Kan? Or the authors of this current manuscript – please clarify who these authors are

Line 111/112 – is the presence of large rubble clasts covering the bottom of the grooves? Does it build up in these grooves for example? Or is it a fine veneer?

Figure 1 – Very nice figure, but its not clear what TD and TS stand for (depression and storm?). Please include in the caption, and also add (OTR) to the caption after One Tree Reef (change from Island to match figure)

Line 126 – add month of bleaching

Line 161 – ‘reef building is more than coral growth’ – this needs further explanation. I think you need to explain the role that rubble plays in reef building as well

Figure 2 – Great pictures – but do you have an equivalent picture to D) that you could use for C) instead or as well? C shows the SAG clearly but does not show the landward section of reef so you cannot get an idea of how ‘new’ these rubble fans are (are they all rubble, or a mixture of sand too?) . The description of E) needs some additional information – this is a combo of LiDAR from 2018? Plus orthomosaic? Its not clear what data are from 2018 and what are from 2025?. In (G) suggest changing key to LiDAR bathy and drone bathy. It would be interesting to overlay the location of grooves upon E) to determine if the location of these fans are directly landward or slightly angled relative to the centre of the grooves (does one groove produce one fan, or multiple grooves?).

---

## [Reviewer Report]

This is an exceptionally valuable report on phenomena occurring at the reef edge of coral reefs during heavy swells, based on data and observations from the period around intense, infrequent tropical cyclones. Unlike sand cays, the seafloor where coral rubble accumulates and the islets composed of coral rubble are difficult to observe continuously as they change. Observations during major events like Tropical Cyclone Alfred provide a realistic and reliable opportunity to obtain evidence. The authors' initiative in successfully seizing this extremely valuable chance is commendable.

The text is concise yet contains important points and is commendable. However, Figure 2 contains many unclear points and should be improved.

Figure 2

1) Is it possible to describe where changes occurred by comparing (A) and (B)?

For example, the shape of the spite on the reef northeast of OTI has also changed (the elongated spite seaward has disappeared, and sedimentation is progressing near the island). The vegetated spite northwest has become narrower (has erosion progressed?).

2) Are (C) and (D) photographs from the same viewpoint? (C) appears to be a vertical photograph, while (D) seems to be an oblique photograph.

It is unclear what “the grooves off the SE of One Tree Reef” refers to; please indicate this with arrow(s) on the figure or add an explanatory note.

Please also add arrow(s) to photograph (D) for the description “infilled with coral rubble fans spilling sediment onto the reef flat”.

Regarding wave conditions at the reef edge during tropical cyclones, there is a paper by Watanabe et al. 2023.

https://doi.org/10.1016/j.oceaneng.2023.115632

Note that the filling of grooves with coral gravel is also evident from observations of cross-sections of cutting through coral reefs (Kan et al. 1995, 1997). These papers are for your information. They can be viewed at the following sites.

Kan, H., Hori, N., Nakashima,Y. and Ichikawa,K. (1995) The evolution of narrow reef flats at high-latitude in the Ryukyu Islands. Coral Reefs, Vol.14, p.123-130.

https://link.springer.com/article/10.1007/BF00367229

Kan,H., Hori,N., Kawana,T., Kaigara,T. and Ichikawa,K. (1997) The evolution of a Holocene fringing reef and island: reefal environmental sequence and sea level change in Tonaki Island, the central Ryukyus. Atoll Research Bulletin, No.443, p.1-20.

https://d1wqtxts1xzle7.cloudfront.net/84708123/Kan_et_al_1997_Atoll_Res_Bull_Smithsonian_-libre.pdf?1650678660=&response-content-disposition=inline%3B+filename%3DThe_evolution_of_a_Holocene_fringing_ree.pdf&Expires=1758677739&Signature=Uvej7851~vkFT59z~0CisQaRbadJd0~8n1IG7jTapPqo1udvy6G-eU0LuYXVGL0xnUMvypX6FUluUOCLAcYhLD7DFsCVw~DLdMKfG1PHKKKvUyNE-Oc7JcKifog2bpHUC04TwUWv0mBEfds3lXaBIUaWrNmjExCU0CkGof~UEjKqbZH4TVQXfQV~WHk94S0D~dfn90clRP9-M3-Gm4HAUXap8-XGRM-rnNV92VCStnEZQYfEj5fwgtik2NMtbu1cJ19iFCGjszHN-xHueRjrpGXdy5-2NndYParJJ~RsFTUyDLfUT6Kf5x9uJujJ-v8oavg7SBHp2AEr4QBBcmJ4zQ__&Key-Pair-Id=APKAJLOHF5GGSLRBV4ZA

Since I can’t seem to attach the file, I’ll provide the website instead.

---

## [Editor Report]

The two reviewers provide very positive examinations of your manuscript. I agree that this work should be published with some minor / moderate revisions. These all seem relatively straightforward and think this team of authors will have no problem making the necessary revisions.

I do have some queries and confusion about the way this research is framed. It seems as though spur and groove outer reef morphologies are being described as an avenue for sediment transport for building reef islands. My understanding of reef islands comes from the GBR / Seychelles / Malaysia and for those sites, reef islands tend to build up on the leeward, rather than the windward, side of a reef platform. There are no SaGs on the leeward side of these platforms.

The study site of this paper, One Tree Island, is a shingle (rather than sand), island and a rather anomalous case for the Capricorn Bunker Group, and indeed the wider GBR as the island has formed on the windward side of the reef. To make a direct linkage between the SnG sediment stores and islands, and to claim that this work has significance for low lying island communities elsewhere seems a stretch. 

The authors should either reframe, or explicitly name some reef islands (i.e. sand cays of the sort that may be populated in the Pacific, where islanders are seeing the impacts of sea-level rise) that have actually formed or been nourished through sediment transport adjacent to spur and grooves. There has been a tendency in recent years to identify sediment in any part of the reef system as an immediate saviour of reef islands. Most have no credible foundation. 

Funafuti could be an example where windward islands have formed through coral rubble accumulation. The reef flats are narrow (<100m). While there is little doubt the coral rubble must have been derived from the forereef (which does have SnG), the mechanisms could be storm removal of coral and gravel along the entire forereef as opposed to SnG being the conveyor.

A broader question is why they make this link? The work is significant without leveraging off the discussion around island susceptibility to change, and is this necessary? Perhaps the article could be framed around the storage and potential flux of gravel to reef flats more generally?

Finally, in the acknowledgements section you acknowledge the first nations TOs for OTI, can you name and/ or list the groups?

---

## [Editor Report]

This manuscript has been much improved by the revisions - I am happy to accept without it going out to a further review, subject to a few minor amendments, as outlined in the attached version.

Well done to this research team for this valuable paper!